# Enzyme-Assisted Method for Phycobiliproteins Extraction from *Porphyra* and Evaluation of Their Bioactivity

**Chung-Hsiung Huang** *, **Wei-Chen Chen, Yu-Huei Gao, Guan-Wen Chen** , **Hong-Ting Victor Lin** and **Chorng-Liang Pan** *

Department of Food Science, National Taiwan Ocean University, Keelung 20224, Taiwan; s1020094@gm.pu.edu.tw (W.-C.C.); ab123k6@gmail.com (Y.-H.G.); chengw@mail.ntou.edu.tw (G.-W.C.); HL358@mail.ntou.edu.tw (H.-T.V.L.)

* Correspondence: huangch@mail.ntou.edu.tw (C.-H.H.); b0037@mail.ntou.edu.tw (C.-L.P.);
  Tel.: +886-2-2462-2192 (ext. 5115) (C.-H.H.); +886-2-2462-2192 (ext. 5116) (C.-L.P.)

**Abstract:** Due to the poor protein availability of algae in their unprocessed form, development of extraction methods for phycobiliproteins is of great significance. This study aimed to extract phycoerythrin (PE) and phycocyanin (PC) from *Porphyra* via bacterial enzymatic hydrolysis and to evaluate their bioactivity. To induce enzyme production, *Porphyra* powder was added into the culture medium of two marine bacterial strains. The pH and enzyme activity of the cultured supernatant, namely crude enzyme solution, were significantly raised. For PE and PC extraction, *Porphyra* were incubated within crude enzyme solution with homogenization and ultrasonication followed by ultrafiltration process. After distinguishing by fast performance liquid chromatography (FPLC), three major fractions were observed and identified as R-PE, R-PC and small molecular PE by high-performance liquid chromatography (HPLC) and polyacrylamide gel electrophoresis (PAGE) analysis. With respect to bioactivity, these three fractions exhibited free radical scavenging and antioxidant activities in a various degree. In addition, the angiotensin-converting-enzyme (ACE) inhibitory activity of both R-PE and R-PC fractions was observed in a concentration-dependent manner. Taken together, the employed process of bacterial enzymatic hydrolysis is suggested to be a feasible method to obtain PE and PC from *Porphyra* without limiting their bioactivity.

**Keywords:** angiotensin-converting-enzyme inhibitor; antioxidant; bacterial enzymatic hydrolysis; phycoerythrin; phycocyanin; *Porphyra*

## 1. Introduction

Algae biomass is considered a novel source of protein and is rich in carbohydrates. Compared with terrestrial crops, algae have the advantages of rapid growth, high carbon fixation ability, higher protein yield per unit area, simple cultivation processes, low breeding cost without land occupation and fresh water consumption [1]. *Porphyra*, one of the most cultured red algae, is gaining in economic importance, and is cultivated, harvested, dried, processed and consumed in large quantities in East Asia. Compared with other algaes, *Porphyra* is rich in protein with a protein content of 7–50%, and about 75% of that is digestible [1].

Phycobiliproteins, one of the compounds most abundant in *Porphyra*, exhibit various biological activities, including immunomodulating, anticancer, antihyperlipidemic, antioxidative and angiotensin-converting-enzyme (ACE) inhibitory effects [2–5]. Phycobiliproteins, commonly composed of α, β and γ subunits, are a family of light-harvesting pigment-protein complexes [6]. The absorption maxima of R-PE are at 498 and 565 nm, with an absorption shoulder at 540 nm, and those of R-PC are at 550 nm and 615 nm [7,8]. In addition, both R-PE and R-PC are widely used for imaging studies owing to their fluorescence properties [1]. However, widespread use of algae proteins is limited by several factors, including high carbohydrate content, low digestibility in the form of raw algae,

and the availability of scalable production methods for protein extraction [9]. Current processes of algal protein extraction are time-consuming and economically unviable [10]. So far, extraction of algal proteins is a relatively poorly studied topic compared to proteins from other crops

Algal proteins are conventionally extracted by means of aqueous, acidic, and alkaline methods, followed by several rounds of centrifugation and recovery using techniques such as ultrafiltration, precipitation and chromatography [11]. The successful extraction of algal proteins can be greatly influenced by the availability of the protein molecules, which can be substantially hindered by high viscosity and anionic cell-wall carbohydrates, such as carrageenans in red algae [12]. Cell disruption methods and chemical reagents are therefore used in order to improve the efficiency of algal protein extraction. Some examples of conventional methods that are commonly used include mechanical grinding, osmotic shock, ultrasonic treatment and enzyme-assisted hydrolysis [13]. Nevertheless, these methods may also undermine the integrity of extracted algal proteins due to the release of proteases from cytosolic vacuoles [14]. In the current study, the crude enzyme solutions were produced by two marine bacterial strains. By employing the crude enzyme solutions, an enzyme-assisted method for PE and PC extraction from *Porphyra* was developed and optimized. Furthermore, the antioxidant and ACE inhibitory activity of extracted PE and PC fractions was evaluated to understand whether the developed method is feasible for PE and PC extraction without diminishing their bioactivity.

## 2. Materials and Methods

### 2.1. Chemicals and Reagents

All chemicals and reagents were purchased from Sigma-Aldrich Chemical Co. (St. Louis, MO, USA) and Panreac Química SLU (Castellar del Vallès, Barcelona, Spain) unless otherwise stated. Reagents for bacterial culture were purchased from Difco Laboratories (Detroit, MI, USA). *Porphyra* sp., purchased from Xin meng cheng company (Penghu, Taiwan), was ground, sieved (0.38 mm pore size) and stored at 4 °C before use. The proximate composition, including the moisture, crude protein, crude lipid, crude fiber, and ash of the marine algae, were analyzed according to Association of Official Analytical Chemists (AOAC) official methods of analysis [15].

### 2.2. Production of Crude Enzyme Solutions from Marine Bacterial Strains

The bacterial strains of *Pseudomonas vesicularis* MA103 and *Aeromonas salmonicida* MAEF108 were isolated from the seawater off the coast at Keelung in Taiwan as previous described [16]. 0.5 g *Porphyra* powder was added into 100 mL of marine medium broth (MMB) followed by autoclaved to prepare MMB-Porphyra medium. MA103 or MAEF108 ($1 \times 10^6$ CFU/mL) was incubated in MMB-*Porphyra* medium at 26 °C with a shaking rate of 150 rpm. After incubation for 1–5 days, the incubated medium was centrifuged at $12,000 \times g$ for 30 min, and the supernatant was harvested and passed through 0.22 µm filter membrane as crude enzyme solutions for further experiments.

### 2.3. Determination of pH and Enzyme Activity of Crude Enzyme Solutions

The pH value of crude enzyme solutions was determined by a pH meter (CyberScan pH-510, Eutech Instruments, Ayer Rajah Crescent, Singapore). The agarase, carrageenanase, $\alpha$-amylase, cellulase, xylanase and mannanase activities of crude enzyme solutions were measured according to the methods described in previous studies [17–20]. One unit of enzyme activity (U) was defined as the increased in amount of reducing sugar per min measured by absorbance.

### 2.4. Process of Phycobiliprotein Extraction from Porphyra

The process of phycobiliprotein extraction was accomplished according to previous studies with some modifications [21,22]. The samples of dry *Porphyra* or *Porphyra* powder were divided into following groups for process. PorWH: dry *Porphyra* in distilled

water processed with homogenization; PorBH: dry *Porphyra* in sodium-phosphate buffer with homogenization; PorBE: dry *Porphyra* in sodium-phosphate buffer containing 15% MA103 crude enzyme solution and 15% MAEF108 crude enzyme solution; PorBHE: dry *Porphyra* in sodium-phosphate buffer containing 15% MA103 crude enzyme solution and 15% MAEF108 crude enzyme solution processed with homogenization; PorPB: *Porphyra* powder in sodium-phosphate buffer; PorPBE: *Porphyra* powder in sodium-phosphate buffer containing 15% MA103 crude enzyme solution and 15% MAEF108 crude enzyme solution; PorPBHE: *Porphyra* powder in sodium-phosphate buffer containing 15% MA103 crude enzyme solution and 15% MAEF108 crude enzyme solution processed with homogenization; PorPBUE: *Porphyra* powder in sodium-phosphate buffer containing 15% MA103 crude enzyme solution and 15% MAEF108 crude enzyme solution processed with ultrasonication; PorPBHUE: *Porphyra* powder in sodium-phosphate buffer containing 15% MA103 crude enzyme solution and 15% MAEF108 crude enzyme solution processed with homogenization and ultrasonication. In brief, 2 g of dry *Porphyra* or *Porphyra* powder was added in 200 mL of distilled water or sodium-phosphate buffer (pH 7) containing crude enzyme solution (15%; *v/v*). The mixture was homogenized (869-18R, Osterizer Calaxie Cycle Blend 10 Speed, Oster, Boca Raton, FL, USA) for 20 s and ultrasonicated (D150H, Delta New Instruments, New Taipei City, Taiwan) for 10 min followed by incubated at 26 °C for 0–72 h with a shaking speed of 120 rpm. After centrifugation at 12,000× *g* for 20 min, the supernatant was subjected to ultrafiltration (UF) with a 100 kDa hollow fiber membrane (UFP-100-E-6A, GE Healthcare) and passed through 0.22 μm filter membrane. The purity and concentration of PE and PC was determined by detection of fluorescence intensity as the following [23–25].

$$\text{R-PEpurity} = A_{560}/A_{280}$$

$$\text{R-PCpurity} = A_{615}/A_{280}$$

$$\text{R-PEconcentration} = 0.123 \times A_{560} - 0.068 \times A_{615} + 0.015 \times A_{650}$$

$$\text{R-PCconcentration} = 0.162 \times A_{615} - 0.001 \times A_{560} - 0.098 \times A_{650}$$

### 2.5. Purification of Phycobiliproteins by Fast Protein Liquid Chromatography (FPLC)

After UF, the extraction with molecules > 100 kDa (UF > 100 kDa) was purified by ÄKTA pure 25 M system (Cytiva; Little Chalfont, Amersham, UK) along with a SuperdexTM 200 Increase 10/300 GL column with phosphate-buffered saline (PBS) at a flow rate of 0.5 mL/min. The purity of PE and PC for each fraction (0.25 mL) was determined by UV-visible absorbance spectrum. Fractions with the highest purity were harvested for further experiments.

### 2.6. Analysis of PE and PC within Fractions

The amount of PE and PC were analyzed by high-performance liquid chromatography (HPLC) using Asahipak SB-804 HQ (7.5 × 300 mm; Showa Denko America, Inc., New York, NY, USA), along with a UV/Visible detector (280 nm). All the tested samples were filtered through a 0.22 μm membrane filter prior to the HPLC analysis [26]. Furthermore, the fractions were separated by native polyacrylamide gel electrophoresis (Native-PAGE) and sodium dodecyl sulfate polyacrylamide gel electrophoresis (SDS-PAGE) as described in the previous studies [27,28].

### 2.7. Antioxidation and Angiotensin-Converting-Enzyme (ACE) Inhibition Assays

The scavenging activity of 1,1-Diphenyl-2-picrylhydrazyl radical 2,2-Diphenyl-1-(2,4,6-trinitrophenyl)hydrazyl (DPPH) free radical and reducing power were assayed according to the methods of Wang et al. and Shimada et al. [29,30]. Trolox was used as the standard, and free radical scavenging activity and reducing power of tested samples were calculated relative to standard Trolox. $Fe^{2+}$ chelating activity was measured according to the method of Dinis et al. [31]. Ethylenediaminetetraacetic acid (EDTA) was used as the standard,

and $Fe^{2+}$ chelating activity of tested samples were calculated relative to standard EDTA. ACE inhibitory activity of tested samples was measured using ACE Kit-WST (Dojindo EU GmbH; Munich, Germany) according to the manufacturer's instructions.

*2.8. Statistical Analysis*

Data were analyzed statistically using IBM® SPSS Statistic® (IBM, Armonk, NY, USA, 2015). One-way analysis of variance (ANOVA) was used to determine statistical differences between sample means, with the level of significance set at $p < 0.05$. Multiple comparisons of means were done by Duncan's test. All data are expressed as mean $\pm$ standard deviation.

## 3. Results and Discussion

*3.1. Proximate Compositions of Porphyra Powder*

The proximate compositions of dried marine algae *Porphyra* sp. are presented in Table 1. *Porphyra* powder comprised 10.93 $\pm$ 0.41% moisture, 8.66 $\pm$ 1.47% ash, 33.94 $\pm$ 1.59% crude protein, 1.10 $\pm$ 0.45% crude fat, and 45.37 $\pm$ 0.21% carbohydrate. Concordantly, it has been reported that *P. dentata* has crude protein and carbohydrate contents of 30% and 47%, respectively [32]. Moreover, among algae species, red algae is most rich in protein [1]. Accordingly, *Porphyra* is considered a favorable candidate for phycobiliprotein extraction. However, the successful extraction of phycobiliproteins can be substantially hindered by high content of carbohydrates [9].

**Table 1.** The proximity analysis of *Porphyra* powder.

| Components | Content (%) |
|---|---|
| Moisture | 10.93 $\pm$ 0.41 |
| Ash | 8.66 $\pm$ 1.47 |
| Crude protein | 33.94 $\pm$ 1.59 |
| Crude fat | 1.10 $\pm$ 0.45 |
| Carbohydrate | 45.37 $\pm$ 0.21 |

Carbohydrate = 100% − (% of moisture + ash + crude protein + crude fat). Each value is mean $\pm$ standard deviation ($n$ = 3).

*3.2. Enzyme Activity of Crude Enzyme Solutions Produced by Marine Bacterial Strains*

To overcome the obstacle of high carbohydrate content, enzyme solutions with the activities required to digest carbohydrates are necessary for phycobiliprotein extraction from *Porphyra*. Therefore, marine strains of MA103 and MAEF108, which are known to produce agarase, were employed for the production of crude enzyme solution, as described in the Materials and Methods. After incubation of each strain with *Porphyra* powder for 1–5 days, the harvested supernatants exhibited the activities of agarase, amylase, cellulase, xylanase, carrageenanase and mannanase (Figure 1). The pH value of medium from cultured MA103 was slightly increased from day 1 to day 2 (Figure 1A). As the higher activities of agarase, amylase, cellulase, xylanase, carrageenanase and mannanase were observed on day 4, the supernatant of MA 103 cultured for 4 days was collected and employed as crude enzyme solution (Figure 1A). On the other hand, the pH value of medium from cultured MA103 was slightly increased from day 1 to day 5 (Figure 1B). Since the highest agarase and amylase activities were observed on day 2, the supernatant of MAFE 108 cultured for 2 days was collected and employed as crude enzyme solution (Figure 1B).

*3.3. Impact of Solvent, Homogenization and Ultrasonication on Enzyme-Assisted Extraction of Phycobiliproteins from Dry Porphyra and Porphyra Powder*

As mechanical homogenization, osmotic shock, ultrasonic treatment, and enzyme-assisted hydrolysis are conventional disruption methods for the algae cell wall, the impact of solvent, homogenization and ultrasonication on the efficacy of phycobiliprotein extraction was investigated to optimize the process of enzyme-assisted extraction. After the process of extraction, obvious absorption peaks were observed in both dry *Porphyra* and

*Porphyra* powder. However, the level of absorption peaks of processed *Porphyra* powder was significantly higher than that of dry *Porphyra* (Figure 2A), suggesting that greater surface area of powder form could increase the probability of an enzyme-substrate interaction. Compared to the other extraction methods, *Porphyra* powder incubated within sodium-phosphate buffer containing 15% of MA103 and 15% MAEF108 crude enzyme solution (PorPBE) for 24 h showed the highest absorption at the waves of 495 nm, 560 nm and 615 nm (Figure 2A). The highest level of absorption was observed at 24 h compared to that at the other time in the PorPBE (Figure 2B). As the main absorption peaks of PE and PC are at 495 nm, 560 nm and 615 nm, respectively, these results indicate the highest level of PE and PC concentrations was extracted by the process of PorPBE. Therefore, the supernatant of PorPBE at the incubation time of 24 h was harvested for purification.

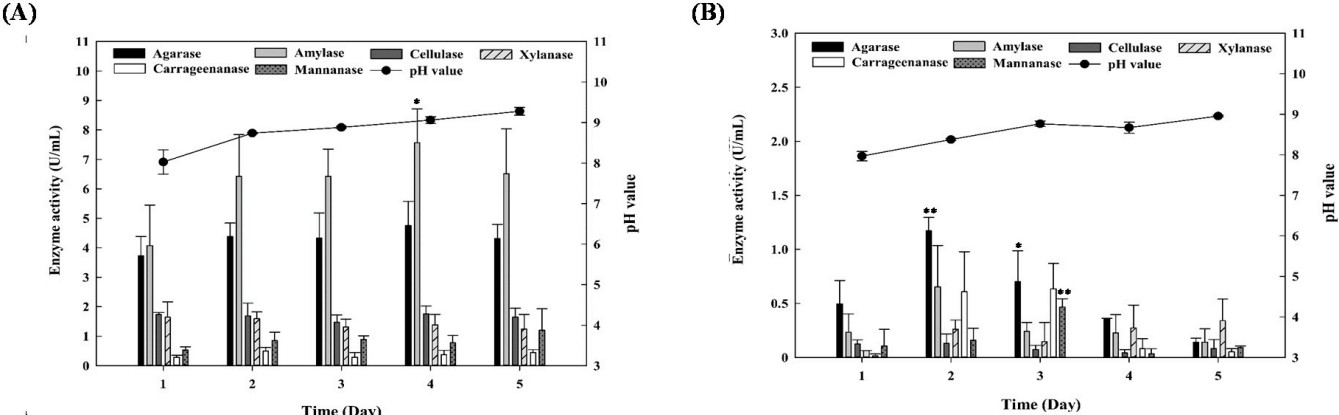

**Figure 1.** The pH value and enzyme activity of crude enzymes solution during incubation period of *P. vesicularis* MA 103 (**A**) and *A. salmonicida* MAEF108 (**B**) induced with MMB-*Porphyra* media. Each value is mean $\pm$ standard deviation ($n = 3$). Comparison of activity of the same enzyme at different time, * and ** indicate significantly different at $p < 0.05$ and $p < 0.01$, respectively.

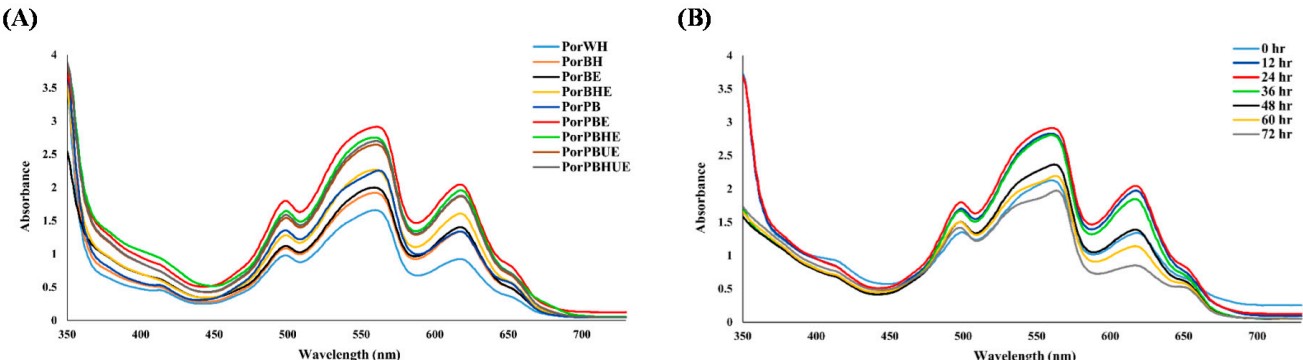

**Figure 2.** UV-visible absorbance spectrum of dry *Porphyra* and *Porphyra* powder extracted by (**A**) different solvents or different assisted extraction methods at 26 °C for 24 h, and (**B**) *Porphyra* powder extracted by sodium-phosphate buffer containing 15% MA103 crude enzyme solution and 15% MAEF108 crude enzyme solution (PorPBE) for 0–72 h. Each line indicates the sample of dry *Porphyra* or *Porphyra* powder extracted by a specific method as described in the Materials and Methods.

### 3.4. Purity of PE and PC

As shown in the Materials and Methods, the supernatant of PorPBE was purified by UF and FPLC. The concentration and purity of R-PE and R-PC in the sample of UF > 100 kDa were greatly higher than that of UF < 100 kDa and that without UF (Table 2). In addition, three major peaks at 560 nm in the FPLC spectrum were observed (Figure 3A). Based on the retention time and molecular weight, the three fractions were identified as R-PE, R-PC

and small molecular PE. Notably, the peak area of R-PE fraction, R-PC fraction and small molecular PE fraction in the spectrum of UF > 100 kDa was about 10 times larger than that without UF (Figure 3B). According to the results determined by UV-Vis spectroscopy, the concentration of PE and PC were about 0.36, 0.86 and 0.13 mg/mL within R-PE fraction, R-PC fraction and small molecular PE fraction, respectively (Table 3). These results indicate that R-PE, R-PC and small molecular PE extracted from *Porphyra* were effectively purified by UF, and the molecular weights of most purified R-PE and R-PC were higher than 100 kDa. Furthermore, the three major fractions were collected and analyzed by HPLC. In agreement with the spectrum of R-PE and C-PC standards, there was only one peak observed in the spectrum of these three fractions (Figure 4), supporting the fact that UF and FPLC purification successfully remove the other proteins.

To further analyze the purified R-PE and R-PC fractions, the UF > 100 kDa and fractons of R-PE and R-PC, as well as R-PE and C-PC standards, were subjected to Native-PAGE. Protein bands with the molecular weight of 240 kDa and that with orange fluorescence under ultraviolet light were observed in the UF > 100 kDa, R-PE fraction and R-PE standard (Figure 5). Protein bands with the molecular weight of 135 kDa and that with red fluorescence under ultraviolet light were observed in the UF > 100 kDa and R-PC fraction (Figure 5). However, the protein band with 125 kDa was observed in the C-PC standard (Figure 5). It has been reported that PE and PC isolated from different algae species show a slight difference in molecular weight [33]. Moreover, Sathuvan et al. demonstrated that PE harvested from *Halymenia floresia* shows orange fluorescence under ultraviolet light [34]. Chethana et al. reported that PC shows red fluorescence under ultraviolet light when it exists in its natural form [35]. Accordingly, protein bands with the molecular weight of 240 and 135 kDa observed in Native-PAGE are suggested as R-PE and R-PC, respectively.

In parallel, these samples were also subjected to SDS-PAGE. Protein bands with various molecular weights were observed in the UF > 100 kDa. Noticeably, protein bands with the molecular weight of 17 and 18 kDa were observed in R-PE standard, C-PC standard and the three fractions; protein bands with the molecular weight of 34 kDa were observed in the R-PE standard and the R- PE fraction (Figure 6). Malairaj et al. pointed out that the molecular weight of α, β, and γ subunits of PE isolated from *Halymenia floresia* are 16, 21 and 39 kDa, respectively [34]. Nair et al. demonstrated that the molecular weight of α, β, and γ subunits of PE harvested from *Centroceras clavulatum* are 18, 19 and 35 kDa, respectively, and that of α and β subunits of PC are 17 and 21 kDa [33]. In addition, Dumay and Morançais indicated that R-PE is a 240 kDa protein with the composition of $(\alpha\beta)_6\gamma$ [6]. Based on the results of PAGE, it is suggested that the R-PE fraction of contained PE had the form of $(\alpha\beta)_6\gamma$. The fraction of small molecular PE was only composed of $(\alpha\beta)_3$, which was possibly due to the breakage of peptide bonds under the pressure of purification process.

**Table 2.** Concentrations, fluorescence intensity and purity indexes of R-PE and R-PC before and after ultrafiltration (100 kDa) of supernatant of PorPBE.

| Sample | R-PE (mg/mL) | R-PC (mg/mL) | Fluorescence Intensity | | Purity Indexes | |
|---|---|---|---|---|---|---|
| | | | R-PE | R-PE | $A_{560}/A_{280}$ | $A_{615}/A_{280}$ |
| without UF | 0.17 ± 0.1 [b] | 0.22 ± 0.1 [b] | 18,505 ± 1144 [a] | 2476 ± 106 [a] | 0.33 | 0.23 |
| UF < 100 kDa | 0.04 ± 0.1 [c] | 0.04 ± 0.1 [c] | 8653 ± 719 [b] | 782 ± 71 [b] | 0.05 | 0.01 |
| UF > 100 kDa | 1.15 ± 0.1 [a] | 0.69 ± 0.1 [a] | 18560 ± 381 [a] | 2492 ± 26 [a] | 0.99 | 0.70 |

Each value is mean ± standard deviation (*n* = 3). Values in the same column with different superscripts are significantly different at $p < 0.05$.

**Table 3.** FPLC analysis of R-PE, R-PC and small molecule PE fractions.

| | PE (mg/mL) | PC (mg/mL) | Fluorescence Intensity | | Purity Indexes | |
|---|---|---|---|---|---|---|
| | | | PE | PC | $A_{560}/A_{280}$ | $A_{615}/A_{280}$ |
| R-PE | $0.36 \pm 0.01$ | $0.03 \pm 0.01$ | $45,438 \pm 2164$ | $1628 \pm 356$ | 3.18 | 0.32 |
| R-PC | $0.06 \pm 0.01$ | $0.86 \pm 0.05$ | $2684 \pm 235$ | $3916 \pm 377$ | 1.75 | 3.37 |
| Small molecule PE | $0.13 \pm 0.02$ | $0.03 \pm 0.01$ | $39,558 \pm 573$ | $950 \pm 357$ | 2.32 | 0.73 |

Each value is mean $\pm$ standard deviation ($n = 3$).

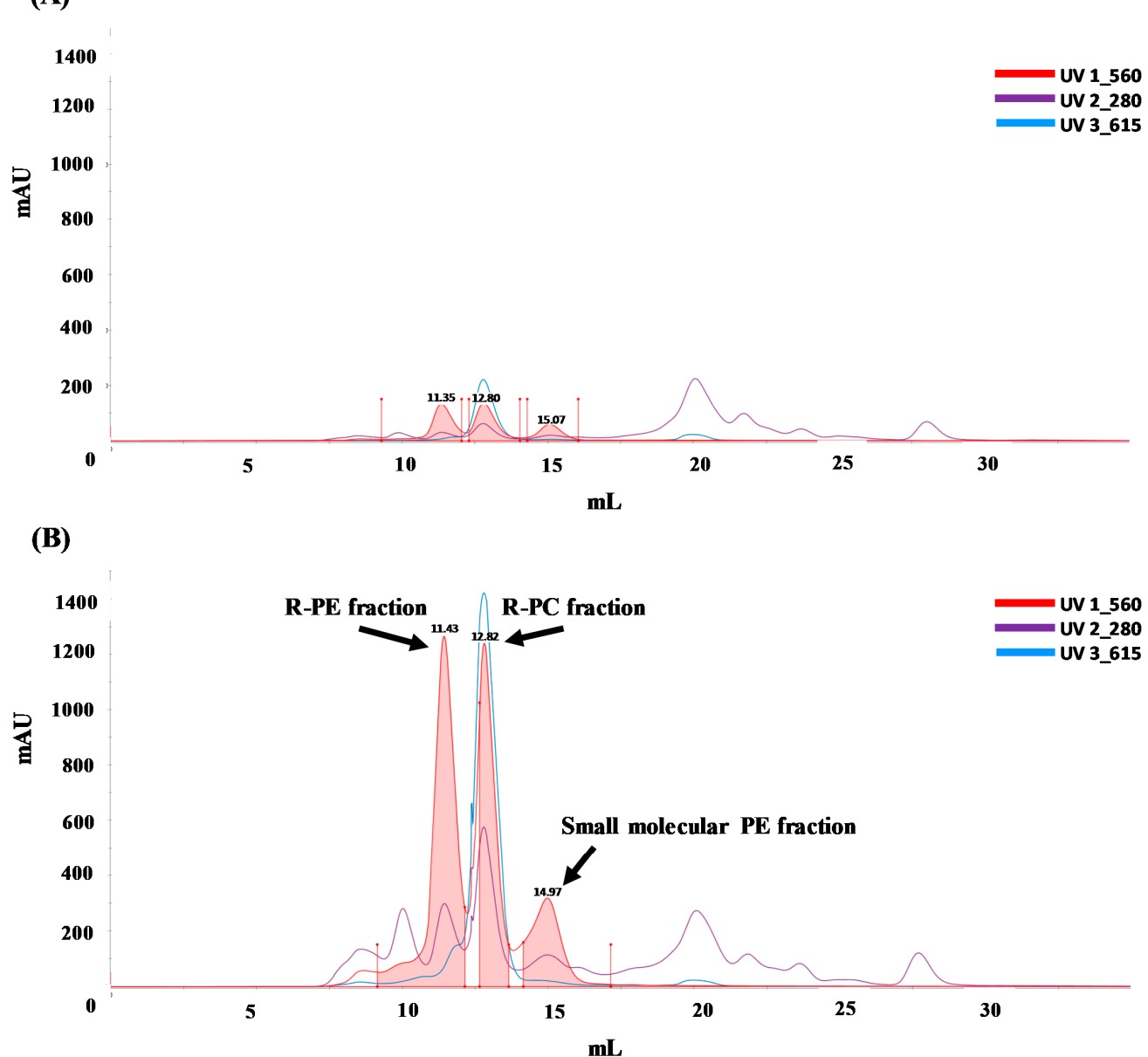

**Figure 3.** FPLC chromatograms of (**A**) unpurified and (**B**) purified *Porphyra* PorPBE extract by ultrafiltration.

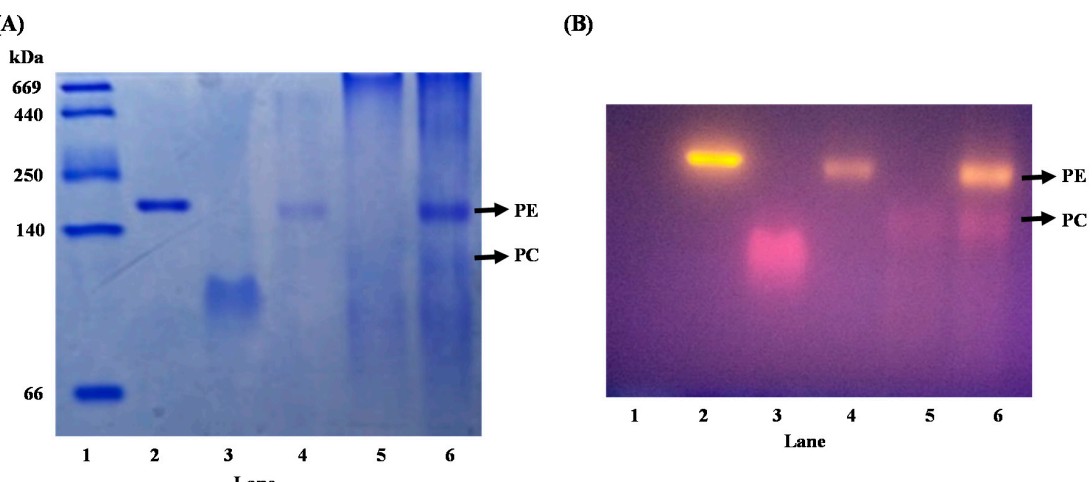

**Figure 4.** HPLC chromatogram of (**A**) R-PE standard, (**B**) C-PC standard, (**C**) R-PE fraction, (**D**) R-PC fraction and (**E**) small molecular PE fraction.

**Figure 5.** Native-PAGE chromatogram of phycobiliproteins from *Porphyra* under (**A**) visible light and (**B**) ultraviolet light. Lane 1: Protein marker; Lane 2: R-PE standard; Lane 3: C-PC standard; Lane 4: R-PE fraction from FPLC; Lane 5: R-PC fraction from FPLC; Lane 6: UF > 100 kDa.

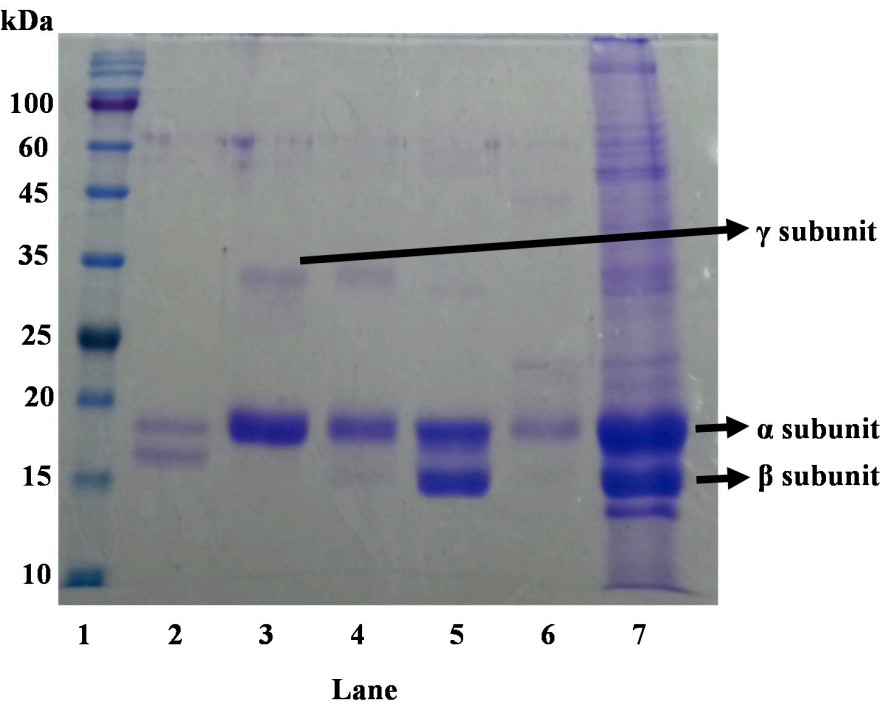

**Figure 6.** SDS-PAGE chromatogram of phycobiliproteins from *Porphyra*. Lane 1: Protein marker; Lane 2: C-PC standard; Lane 3: R-PE standard; Lane 4: R-PE fraction; Lane 5: R-PC fraction; Lane 6: small molecular PE fraction; Lane 7: UF > 100 kDa.

### 3.5. Antioxidant and ACE Inhibitory Activities of Purified PE and PC

To corroborate the bioactivities of purified fractions, their antioxidant and ACE inhibitory activities were explored. The DPPH free radical scavenging, reducing power and ion chelation effects of these three fractions were observed in a concentration-dependent manner (Tables 2–4). The DPPH free radical scavenging rates of R-PE, R-PC and small molecule PE fractions (10 mg/mL) were equivalent to 15.14, 14.26 and 15.36 μg/mL of Trolox (Table 4). The reducing power of these three fractions (10 mg/mL) were equivalent to 11.96, 36.49 and 7.56 μg/mL of Trolox (Table 5). The $Fe^{2+}$ ion chelation rates of these three fractions (2.5 mg/mL) were 91.21%, 90.31%, and 92.11% and equivalent to 0.0632, 0.0621 and 0.0624 mg/mL of EDTA, respectively (Table 6). In addition, both R-PE and R-PC fractions exhibited ACE inhibitory effects in a concentration-dependent manner (Figure 7). ACE inhibitory activity of undiluted R-PE and R-PC fractions were 54.8% and 50.8%, respectively (Figure 7). Taken together, these results indicate that bioactivities of PE and PC were not lost by the processes of extraction and purification.

**Table 4.** DPPH free radicals scavenging effect of phycobiliprotein extracts from *Porphyra*.

| Fractions (mg/mL) | DPPH Scavenging Effect (%) | Equal to μg/mL Trolox |
|---|---|---|
| R-PE | | |
| 5 | 30.17 ± 0.95 [b] | 7.14 ± 0.33 |
| 10 | 63.06 ± 1.08 [a] | 15.14 ± 0.37 |
| R-PC | | |
| 5 | 23.87 ± 1.12 [c] | 5.61 ± 0.39 |
| 10 | 59.46 ± 1.23 [a] | 14.26 ± 0.42 |
| Small molecule PE | | |
| 5 | 28.23 ± 0.88 [b] | 6.67 ± 0.30 |
| 10 | 63.96 ± 1.37 [a] | 15.36 ± 0.47 |

Each value is mean ± standard deviation (*n* = 3). Values in the same column with different superscripts are significantly different at $p < 0.05$.

**Table 5.** Reducing power of phycobiliprotein extracts from *Porphyra*.

| Fractions (mg/mL) | OD$_{700}$nm | Equal to μg/mL Trolox |
|---|---|---|
| R-PE | | |
| 5 | 0.046 ± 0.005 [e] | 5.7493 ± 0.5274 |
| 10 | 0.087 ± 0.007 [c] | 11.958 ± 0.9880 |
| R-PC | | |
| 5 | 0.138 ± 0.006 [b] | 19.5375 ± 0.6028 |
| 10 | 0.250 ± 0.028 [a] | 36.4900 ± 4.2910 |
| Small molecule PE | | |
| 5 | 0.029 ± 0.006 [f] | 3.1123 ± 0.6781 |
| 10 | 0.058 ± 0.005 [d] | 7.5580 ± 0.6910 |

Each value is mean ± standard deviation ($n = 3$). Values in the same column with different superscripts are significantly different at $p < 0.05$.

**Table 6.** Ferrous ion chelating activity of phycobiliprotein extracts from *Porphyra*.

| Fractions (μg/mL) | Fe$^{2+}$ Chelating Effect (%) | Equal to μg/mL EDTA |
|---|---|---|
| R-PE | | |
| 31.25 | 60.59 ± 0.74 [e] | 26.4 ± 0.9 |
| 62.50 | 71.61 ± 0.52 [c] | 39.6 ± 0.6 |
| 125 | 87.63 ± 0.23 [b] | 58.9 ± 0.3 |
| 250 | 91.21 ± 0.11 [a] | 63.2 ± 0.1 |
| R-PC | | |
| 31.25 | 67.10 ± 0.45 [d] | 34.2 ± 0.5 |
| 62.5 | 73.61 ± 0.31 [c] | 42. ± 0.4 |
| 125 | 86.18 ± 0.30 [b] | 57.1 ± 0.4 |
| 250 | 90.31 ± 0.11 [a] | 62.1 ± 0.1 |
| Small molecule PE | | |
| 31.25 | 63.09 ± 1.21 [e] | 29.4 ± 1.5 |
| 62.5 | 73.96 ± 0.23 [c] | 42.5 ± 0.3 |
| 125 | 89.18 ± 0.15 [b] | 60.7 ± 0.2 |
| 250 | 92.11 ± 0.11 [a] | 64.2 ± 0.1 |

Each value is mean ± standard deviation ($n = 3$). Values in the same column with different superscripts are significantly different at $p < 0.05$.

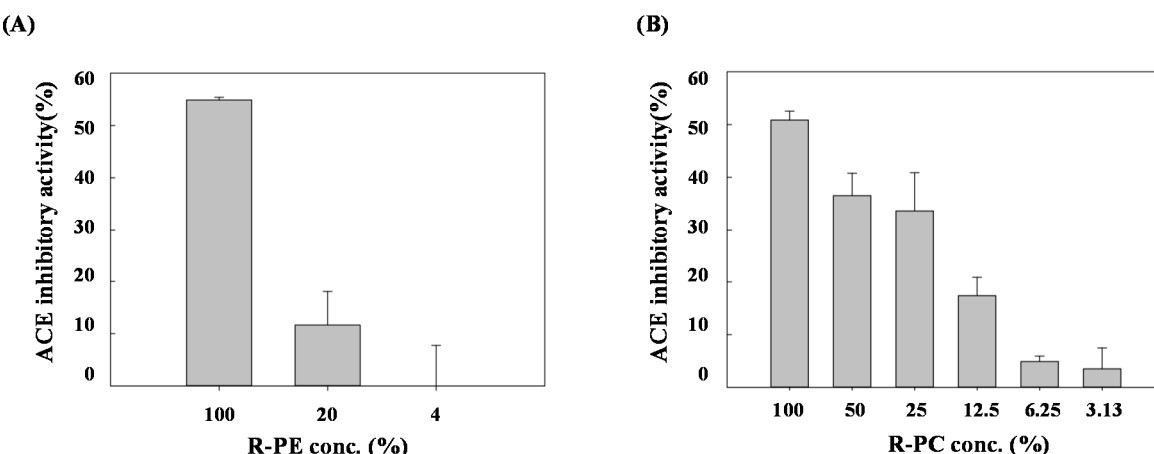

**Figure 7.** Angiotensin converting enzyme (ACE) inhibitory activity1 of (**A**) R-PE and (**B**) R-PC fractions. Deionized water was employed as the control group (without ACE inhibition).

## 4. Conclusions

Red algae are known as a valuable source of bioactive proteins. However, it is necessary to develop a novel strategy to efficiently extract proteins from algae owing to their property of high carbohydrate content. Herein, we firstly induced crude enzyme solution production from marine bacterial strains, and the process of enzyme-assisted extraction of phycobiliproteins from *Porphyra* was optimized. Most importantly, high purity and bioactivity of phycobiliproteins were obtained following UF and FPLC purification, indicating a promising process for PE and PC extraction without depriving their activities. Although further efforts in the achievement of scale-up process are required, this study provided a practical approach for algae protein extraction.

**Author Contributions:** Conceptualization, methodology, investigation, writing—original draft preparation, and review and editing: W.-C.C. and Y.-H.G.; formal analysis, investigation, and editing: G.-W.C. and H.-T.V.L.; review and editing, supervision, project administration, and funding acquisition: C.-H.H. and C.-L.P. All authors have read and agreed to the published version of the manuscript.

**Funding:** This research was supported by the grants from the Ministry of Science and Technology (MOST 108-2221-E-019-039-MY2 and MOST 109-2221-E-019-032-MY3).

**Institutional Review Board Statement:** Not applicable.

**Informed Consent Statement:** Not applicable.

**Data Availability Statement:** Data sharing not applicable.

**Conflicts of Interest:** The authors declare no conflict of interest.

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
