# Peer review of "Enzyme-Assisted Method for Phycobiliproteins Extraction from Porphyra and Evaluation of Their Bioactivity"

_processes, doi:10.3390/pr9030560_

Round 1

Reviewer 1 Report

The study study aimed to extract phycoerythrin and phycocyanin from Porphyra   via  bacterial enzymatic hydrolysis and to further evaluate  their  bioactivity. The research design is appropriate, the  methods are well described and the results are interesting. A significant amount of work has been involved in the study.

Author Response

Response to the Reviewers' comments:

Following is a detailed point-by-point explanation on how we have addressed all the Reviewers’ concerns. The Reviewer’s comments are in black, and our responses are in blue.

Reviewer 1: The study aimed to extract phycoerythrin and phycocyanin from Porphyra via bacterial enzymatic hydrolysis and to further evaluate their bioactivity. The research design is appropriate, the methods are well described and the results are interesting. A significant amount of work has been involved in the study.

˙The authors sincerely appreciate the Reviewer for his/her positive comments.

Reviewer 2 Report

Dear Authors

Please find below my recommendations:

The methods of analysis were presented, but some results are not found in the paragraph dedicated for them:

  1. Concentrations and purities of PE and PC determined by fluorescence
  2.  The purities obtained from the FPLC analysis were determined by UV-Vis spectroscopy, but these results do not appear.
  3.  Regarding the HPLC method of PE analysis and PC there is a discrepancy with the results. The described method is HPLC with a refractive index detector, the indicated bibliographic source refers to HPAEC with Pulse Amperometric Detection. The chromatograms presented in the Results indicate the HPAEC analysis.
  4.  In paragraph 3.1.,the results of the analyzes of crude protein, moisture, ash, crude fat are presented, but it is not specified exactly which are these methods.
  5. In paragraph 3.3. it would be interesting to have a brief discussion regarding the differences between dry and powder samples, in the context of the impact of solvents, homogenization and ultrasonication on PE and PC extraction.
  6.  R-PE fraction and R-PC fraction should be specified in line 201, not to confuse with standard R-PE, but especially with standard C-PC. Or it could be named the PE standard, C-PE ?

Author Response

Response to the Reviewers' comments:

Following is a detailed point-by-point explanation on how we have addressed all the Reviewers’ concerns. The Reviewer’s comments are in black, and our responses are in blue.

Reviewer 2: The methods of analysis were presented, but some results are not found in the paragraph dedicated for them:

˙The authors thank all the Reviewer for his/her appreciation of the importance of the major findings in this study. Following is a detailed point-by-point explanation.

  1. Concentrations and purities of PE and PC determined by fluorescence.

˙The concentrations, fluorescence intensity and purity indexes of R-PE and R-PC before and after ultrafiltration (100 kDa) of supernatant of PorPBE are provided in the revised manuscript (line 217-220, Table 2). The concentration and purity of R-PE and R-PC in the sample of UF > 100 kDa were greatly higher than that of UF < 100 kDa and that without UF (line 203-204).

  1. The purities obtained from the FPLC analysis were determined by UV-Vis spectroscopy, but these results do not appear.

˙FPLC analysis of R-PE, R-PC and small molecule PE fractions is provided in the revised manuscript (line 226-227, Table 3). According to the results determined by UV-Vis spectroscopy, the concentration of PE and PC were about 0.36, 0.86 and 0.13 mg/mL within R-PE, R-PC and small molecular PE fractions, respectively (Table 3). Based on the results shown in Table 2, Table 3 and Figure 3, R-PE, R-PC and small molecular PE extracted from Porphyra were effectively purified by UF, and the molecular weights of most purified R-PE and R-PC were higher than 100 kDa.

  1. Regarding the HPLC method of PE analysis and PC there is a discrepancy with the results. The described method is HPLC with a refractive index detector, the indicated bibliographic source refers to HPAEC with Pulse Amperometric Detection. The chromatograms presented in the Results indicate the HPAEC analysis.

Ë™The authors thank the Reviewer for pointing out the important issue. "Refractive index detector" is corrected as UV/Visible detector (280 nm) in the revised manuscript (line 126).  

  1. In paragraph 3.1.,the results of the analyzes of crude protein, moisture, ash, crude fat are presented, but it is not specified exactly which are these methods.

˙The authors thank the Reviewer for pointing out the important issue. The proximate composition, including the moisture, crude protein, crude lipid, crude fiber, and ash of the marine algae, were analyzed according to Association of Official Analytical Chemists (AOAC) official methods of analysis [15] (line 72-74).

  1. In paragraph 3.3. it would be interesting to have a brief discussion regarding the differences between dry and powder samples, in the context of the impact of solvents, homogenization and ultrasonication on PE and PC extraction.

˙The authors thank the Reviewer for his/her valuable suggestion. A brief discussion is provided in the revised manuscript. After the process of extraction, obvious absorption peaks were observed in both dry Porphyra and Porphyra powder. However, the level of absorption peaks of processed Porphyra powder was significantly higher than that of dry Porphyra (Figure 2A), suggesting that greater surface area of powder form could increase the probability of an enzyme-substrate interaction (line 183-187).

  1. R-PE fraction and R-PC fraction should be specified in line 201, not to confuse with standard R-PE, but especially with standard C-PC. Or it could be named the PE standard, C-PE ?

˙The authors thank the Reviewer for pointing out the important issue. R-PE fraction, R-PC fraction and small molecular PE fraction are specified in the revised manuscript (line 206-210).
